# Targeting Ubiquitin-like Protein, ISG15, as a Novel Tumor Associated Antigen in Colorectal Cancer

**DOI:** 10.3390/cancers15041237

**Published:** 2023-02-15

**Authors:** Hong-My Nguyen, Shreyas Gaikwad, Mariam Oladejo, Wyatt Paulishak, Laurence M. Wood

**Affiliations:** Department of Immunotherapeutics and Biotechnology, Jerry H Hodge School of Pharmacy, Texas Tech University Health Sciences Center, Abilene, TX 79601, USA

**Keywords:** interferon-stimulated gene 15, Listeria-based vaccines, colorectal cancer

## Abstract

**Simple Summary:**

Despite their potential and promising anti-tumor efficacy, previously developed cancer vaccines targeting different tumor-associated antigens for colorectal cancer (CRC) have not yet proven to be successful. Interferon-stimulated gene 15 (ISG15) is emerging as an important oncoprotein with a potential diagnostic signature and therapeutic target for a multitude of cancer, including CRC. In this study, we investigated the therapeutic efficacy of Listeria-based vaccines targeting ISG15 (Lm-LLO-ISG15) in CRC. We found that the Lm-LLO-ISG15 vaccination results in anti-tumor efficacy against the MC38 tumor model by recruiting cytotoxic T lymphocytes and leading to a more favorable effector to regulatory T cell ratio (T_eff_/T_reg_) in the tumor microenvironment.

**Abstract:**

Colorectal cancer (CRC) is the third leading cause of cancer-related deaths in both men and women in the United States. While immune checkpoint inhibitor (ICI) therapy is demonstrating remarkable clinical responses, the resistance and immune-related toxicities associated with ICIs demonstrate the need to develop additional immunotherapy options for CRC patients. Cancer vaccines represent a safe and promising treatment approach for CRC. As previously developed tumor-associated antigen (TAA)-based cancer vaccines for CRC are not demonstrating promising results, we propose that interferon-stimulated gene 15 (ISG15) is a novel TAA and therapeutic target for CRC. Our work demonstrates the anti-tumor efficacy of a Listeria-based vaccine targeting ISG15, designated Lm-LLO-ISG15, in an immunocompetent CRC murine model. The Lm-LLO-ISG15-mediated anti-tumor response is associated with an increased influx of functional T cells, higher production of multiple intracellular cytokines response, a lower number of regulatory T cells, and a greater ratio of effector to regulatory T cells (T_eff_/T_reg_) in the tumor microenvironment.

## 1. Introduction

Colorectal cancer (CRC) is the third leading cause of cancer-related deaths in both men and women in the United States [1,2]. There is an urgent need to develop new treatment regimen(s) for advanced CRC patients with distant metastasis as (i) their current five-year survival rates are only 15% [1,2] and (ii) the initially effective systemic therapies eventually become ineffective due to the development of resistance and intolerable toxicities [3,4,5,6]. Immunotherapy is an emerging treatment approach for CRC with immune checkpoint inhibitors (ICIs), including anti-PD-1/-PD-L1 and anti-CTLA-4 antibodies, demonstrating remarkable clinical responses [7,8,9]. However, the primary/acquired resistance and immune-related toxicities associated with ICIs demonstrate the need to develop additional immunotherapy options for CRC patients [9,10,11].

Cancer vaccines have been demonstrated as a potential platform of immunotherapy for the treatment of CRC [12]. Listeria-based vaccines, an active form of cancer immunotherapy that has demonstrated promising anti-tumor efficacy in different cancers, have, to date, been modestly investigated in CRC [13,14,15]. Listeria-based vaccines are developed from an attenuated *Listeria monocytogenes* (*Lm*) bacterium that has the unique ability to preferentially infect antigen-presenting cells, such as dendritic cells (DCs), to deliver the antigen of interest and induce robust cytotoxic T lymphocyte (CTL) responses that eradicate tumors [13]. Therefore, one of the most important features that determine the therapeutic efficacy of *Lm*-based vaccines is the choice of the antigen. The previously-developed tumor-associated antigen (TAA)-based cancer vaccines for CRC, targeting CEA, MAGE, and GUCY2C, are not currently demonstrating promising results [16]. Therefore, it is essential to discover novel antigens for CRC that can be effectively targeted to deliver significant therapeutic benefits for patients.

ISG15 is a small ubiquitin-like protein regulated by type-I interferon (IFN) and has been known for its essential contribution to the host’s defense mechanism against intracellular pathogens [17]. Interestingly, ISG15 is emerging as an important oncoprotein with a potential diagnostic signature and therapeutic target for several cancers in recent years [18,19,20]. However, how ISG15 expression correlates with CRC progression and survival has not yet been evaluated. In this manuscript, we sought to determine whether ISG15 could be employed as a TAA in CRC. Further, we also evaluated the anti-tumor effect of Lm-LLO-ISG15, a *Lm*-based vaccine targeting ISG15. Our work demonstrated the therapeutic efficacy of Lm-LLO-ISG15 in both subcutaneous and orthotopic syngeneic CRC mouse models.

## 2. Materials and Methods

### 2.1. Animals

All C57BL/6 female mice (6–8-week-old) used in the study were received from either Jackson Laboratories or Envigo. Upon arrival, all mice were caged at the animal core facility at the Laboratory Animal Resources Center (LARC) of TTUHSC-Abilene campus. All in vivo studies were conducted in compliance with the regulations of the Institutional Animal Care and Use Committee (IACUC) at TTUHSC.

### 2.2. Cell Lines

The non-transformed NIH-3T3 fibroblast cell line was purchased from American Type Culture Collection (ATCC). The colorectal cancer cell line MC38 was a kind gift from Dr. Devin Lowe, Texas Tech University Health Sciences Center. The MC38 expressing green fluorescent protein and luciferase (MC38-GL) was a kind gift from Dr. Michael D. Green, University of Michigan. The human colorectal cancer cell line HCT116 was a kind gift from Dr. Sanjay Srivastava, Texas Tech University Health Sciences Center. All cell lines were cultured at 37 °C, 5% CO_2_ in 10% heat-inactivated fetal bovine serum (FBS)/RPMI 1640 media, 50 U/mL penicillin, and 50 µg/mL streptomycin. The cells were low-passage and confirmed to be Mycoplasma-free (MycoAlert, Lonza, Walkersville, MD, USA) before conducting experiments.

### 2.3. Listeria Monocytogenes Strains

The engineering and design of the Lm-LLO-ISG15 vaccine was previously described [15]. Briefly, under control of an hly promoter, a sequence of mouse ISG15 (NM_015783) was genetically fused to truncated Listeriolysin O (tLLO) [15]. Lm-LLO-OVA was used as the control vaccine. Instead of ISG15, the Lm-LLO-OVA vaccine has chicken ovalbumin (OVA) (NM_205152) fused to tLLO [15]. All *Lm*-based vaccines were allowed to grow in Brain Heart Infusion (BHI) media consisting of chloramphenicol (25 µg/mL) and streptomycin (34 µg/mL). A colony-formation assay was used to determine the titer of the *Lm*-based vaccines before all mouse experiments. For the intraperitoneally (i.p.) route of administration, the *Lm*-based vaccines (2 × 10^8^ CFUs) were resuspended in 200 µL of Phosphate Buffered Saline (PBS) and delivered to the animals using 1 mL insulin syringe. For the oral route of administration (p.o.), the *Lm*-based vaccines at 2 × 10^9^ CFUs was resuspended in 100 µL of PBS and given to the animal using oral gavage.

### 2.4. ISG15 Expression in Normal and Tumor Mouse Colon Tissue

RNA was extracted from 3T3, MC38, healthy organs, and subcutaneously implanted MC38 tumors using the RNeasy RNA extraction kit (Qiagen). The RNA samples were then converted to cDNA using a High-Capacity Reverse Transcriptase Kit (Applied Biosystems) followed the manufacturer’s instruction. The cDNA were then subjected to qPCR analysis using ISG15 forward primer (5′-ATGGCCTGGGACCTAAAG-3′) and reverse primer (5′-TTAGGCACACTGGTCCCC-3′), 18S rRNA ISG15 expression was interpreted after normalization with 18S ribosomal RNA with forward primer (5′-CGGCTACCACATCCAAG GAA-3′) and reverse primer (5′-GCTGGAATTACCGCGGCT-3′).

### 2.5. Western Blot Analysis

The 3T3, MC38, normal colon tissues, and subcutaneously implanted MC38 tumors were lysed with RIPA buffer supplemented with protease inhibitors. All of the whole lysates were then mixed with LDS sample loading buffer and reducing agent and subjected to gel electrophoresis with a 4–12% Bis-Tris polyacrylamide gel. After separation, the proteins were transferred to a PVDF membrane overnight at 4 °C. The membrane was incubated with anti-ISG15 polyclonal antibody (PA5-79523, Invitrogen) and anti-β-actin rabbit mAb (4970, CST) at 4 °C. The next day, the blots were incubated with peroxidase-conjugated goat anti-rabbit antibody (Invitrogen, #32460) as a secondary antibody for ISG15 and β-actin. Signals were developed with enhanced chemiluminescence substrate (Thermo Scientific, Waltham, MA, USA) and visualized using a UVP imager. All the whole western blot figures can be found in Appendix A.

### 2.6. In Vitro Cytotoxicity and Infectivity Assays

To determine the cytotoxicity of the *Lm*-based vaccines in vitro, 5 × 10^3^ MC38 or HCT116 cells were first seeded into 96-well plates overnight. The next day, the cells were treated with either Lm-LLO-ISG15, or Control Lm at various MOIs [21]. After three hours, the cells were washed and allowed to grow in RPMI 1640 media containing gentamicin (50 µg/mL, to remove remaining Listeria) for 48 h. After 48 h, the cell viability was determined using SRB assay, following the manufacturer’s protocol.

To determine the infectivity of the *Lm*-based vaccines in vitro, 3.25 × 10^4^ MC38 or HCT116 cells were plated into 96-well plates overnight. Subsequently, the cells were infected with either Lm-LLO-ISG15 or Control Lm at different MOI for 3 h. After removing the supernatant, the cells were washed with PBS and subjected to gentamicin (5 µg/mL) for 1 h to remove all *Lm* that remained. Followed by gentamycin treatment, the cells were lysed using water and the titer of *Lm* was determined by a colony-forming assay.

### 2.7. Tumor Immunotherapy with Lm-LLO-ISG15

For subcutaneous studies, 5 × 10^5^ MC38 cells were suspended in 100 µL PBS and implanted subcutaneously (s.c.) into the right hind flank of C57BL/6 female mice. All of the mice were then randomized to receive PBS, Control Lm, or Lm-LLO-ISG15 intraperitoneally. Tumor measurement was carried out every 2–3 days by digital caliper and tumor volume was calculated using the formula (length × width × width)/2. All of the mice were followed for illness until they became moribund, or when tumors reached the burden limit, i.e., 15 mm in either dimension.

For orthotopic studies, 1 × 10^5^ MC38-GL in Matrigel (Corning Inc., Corning, NY, USA) (25 µL of PBS containing 1 × 10^5^ MC38-GL + 25 µL of Matrigel) were implanted into the cecal wall, as previously described [22,23], and the mice were subsequently treated with either PBS, Control Lm, or Lm-LLO-ISG15 i.p. or p.o. This indicated that the PBS group received PBS i.p. and p.o, the Lm-LLO-ISG15 i.p. group received Lm-LLO-ISG15 i.p. and PBS p.o., while the Lm-LLO-ISG15 p.o. group received PBS i.p. and Lm-LLO-ISG15 p.o. D-luciferin (150 mg/kg, GoldBio, St Louis, MO, USA) was prepared and injected i.p. into the mice and bioluminescence signals were detected by an IVIS Imaging system after 10 min post-injection. The tumor kinetic curve was plotted by using total flux (photon/s) of the region of interest (ROI).

### 2.8. Lymphocyte Depletion Experiments

MC38 subcutaneous tumor-bearing mice were intraperitoneally administered with either PBS or anti-CD8α antibody (clone 2.43, 200 µg/injection, BioXCell, Lebanon, NH, USA) on days 1, 3, 6, 9, and 12. To determine the role of CD8^+^ T cells in the anti-tumor efficacy of Lm-LLO-ISG15, three groups of mice were included: Group 1 (Mock) received PBS + PBS, group 2 (Vaccinated) received Lm-LLO-ISG15 + PBS, and group 3 (CD8^+^ T-cell depletion) received Lm-LLO-ISG15 + anti-CD8α. The mice were monitored for tumor growth kinetics and survival.

### 2.9. Multi-Color Flow Cytometry

To analyze the tumor-infiltrating immune cells, tumors were harvested and passed through a 70-μm cell strainer to collect single-cell suspensions and counted using a Vi-CELL XR Cell Viability Analyzer (Beckman Coulter, Brea, CA, USA), as previously described [24]. Between 1–2 million cells were then plated into round bottom 96-well plates and pre-incubated with purified anti-CD16/32 unconjugated antibody (clone 93) to block the Fc receptors prior to surface staining with fluorochrome-conjugated anti-mouse monoclonal antibodies, which include: APC/Cyanine7 TCR-β chain (clone H57-597), PE/Cy7 CD8α (clone 53-6.7), PerCP/Cy5.5 CD4 (clone GK1.5), FITC PD-1 (clone 29F.1A12), PE FOXP3 (clone 150D), PE IFN-γ (clone XMG1.2), AF700 TNF-α (clone MP6-XT22), APC IL-2 (clone JES6-5H4), APC CD11b (clone M1/70), and PerCP/Cy5.5 Gr-1(clone RB6-8C5). Dead cells were determined using Zombie Aqua Dye following the manufacturer’s instructions. Zombie Aqua Dye was diluted in PBS and added to the primary antibodies containing samples after 30 min. Next, 2% paraformaldehyde (PFA) was used to fix the cells and stored at 4 °C in dark until analyzed. To determine IL-2, IFN-γ, and TNF-α secreting T cells, single cells dissociated from tumors were first stimulated with Cell Activation Cocktail kit (423303, Biolegend, San Diego, CA, USA) in a 96-well plate at 37 °C, 5% CO_2_ in serum-free RPMI 1640 media. After 6 h stimulation, the cells were processed as normal samples. Intracellular staining (IFN-γ, IL-2, and TNF-α) was performed using the FOXP3 intracellular staining protocol (eBioscience, San Diego, CA, USA). FOXP3 was stained using the same protocol omitting the stimulation process. UltraComp eBeads (eBioscience, San Diego, CA, USA) were used to prepare single-color compensation controls for each fluorescently conjugated antibody according to manufacturer instructions. Data analysis was performed using FlowJo software version 10.7.0 with the previously described gating strategies [21].

### 2.10. Statistical Analyses

Prism 8 GraphPad software version 9.4.1. was used to determine all of the statistical analysis. For tumor growth kinetics studies and immune cell infiltration analysis, unpaired student *t*-tests were used. For survival studies, median survival was determined using Kaplan-Meier curves. Comparisons between the groups were performed by Log Rank test. *p* values for all statistical comparisons were defined as * *p* < 0.05, ** *p* < 0.01, *** *p* < 0.001, and **** *p* < 0.0001.

## 3. Results

### 3.1. Elevation of ISG15 Expression in CRC Tumors Is Correlated with an Unfavorable Prognosis

The ISG15 levels in human CRC were analyzed using the publicly available data from the Human Protein Atlas, The Cancer Genome Atlas, and The University of Alabama at Birmingham, Comprehensive Cancer Center (UALCAN). ISG15 mRNA expression is found to be low across all healthy tissues (Appendix A) and almost undetectable at the protein level (Appendix A). Primary CRC tumors consistently express a higher level of ISG15 compared to that of normal colon tissues at both the transcriptional (Figure 1A) and translational levels (Figure 1B). The elevated expression of ISG15 is found to be consistent across all cancer stages (Figure 1C), as well as CRC subtypes (Figure 1D), and does not discriminate between males versus females (Figure 1E). Noticeably, ISG15 tends to be higher in Caucasians and Asians, but not African-Americans, than in the overall population (Figure 1F). We observed that the elevated expression of ISG15 in CRC patients is strongly correlated with shorter survival outcomes (Figure 1G). Collectively, our findings indicate that ISG15 can potentially be used as a TAA in CRC.

### 3.2. High Level of ISG15 Expression in Human CRC Is Also Conserved in Murine CRC

As we thought that ISG15 could possibly provide a potential therapeutic target in CRC, we first sought to confirm whether the phenomenon of ISG15 overexpression in CRC tumors is also conserved in mice. We found that MC38, a common murine model of human CRC, expressed a higher level of ISG15 compared to a non-transformed fibroblast cell line, NIH-3T3 (Figure 2A,D,E). In addition, subcutaneously-implanted MC38 tumors also expressed a higher level of ISG15 compared to normal colon tissue (Figure 2B,D,E) and other organs (Appendix A). Followed by 24-h of IFN-β stimulation, ISG15 expression in MC38 is further induced (Figure 2C).

### 3.3. Lm-LLO-ISG15 Demonstrates Potential Anti-Tumor Effects in Subcutaneous CRC Mouse Model in a CD8^+^ T Cell-Dependent Manner

Next, we aimed to evaluate whether the elevated expression of ISG15 could be a therapeutic target for Lm-LLO-ISG15. MC38 cells were implanted subcutaneously (s.c) in the hind flank of female C57BL/6 mice, as mentioned in the Materials and Methods. The tumor-bearing mice were injected with either PBS or Lm-LLO-ISG15 (Figure 3A). The vaccination of Lm-LLO-ISG15 substantially controlled the tumor burden (Figure 3B and Appendix A) led to an extension in the median survival compared to PBS (Figure 3C).

We further investigated how the Lm-LLO-ISG15 vaccination modulated the tumor microenvironment (TME) to exert anti-cancer efficacy. We observed an increased influx of total T cells, as well as CD4^+^ T cell and CD8^+^ T cell subsets, in the Lm-LLO-ISG15 group compared to that of PBS (Figure 3D–F). Further, regulatory T cells (Tregs, CD4^+^FoxP3^+^) tended to be lower in the Lm-LLO-ISG15-treated tumors, although the difference did not reach statistical significance (Figure 3G). However, we did find that the ratio of CD8^+^ T cells/Tregs was strikingly higher in the Lm-LLO-ISG15 group compared to PBS (Figure 3H). We observed an increasing trend in the number of functional CD4^+^ T cells from the mice treated with Lm-LLO-ISG15 compared to CD4^+^ T cells from PBS, although statistical significance was not met (Figure 3I,J and Appendix A). In contrast, treatment with Lm-LLO-ISG15 generates a larger pool of CD8^+^IFN-γ^+^ and CD8^+^IL-2^+^ population compared to that of PBS (Figure 3K,L). As a result, the total single cytokine-producing CD8^+^ T cells was higher in the Lm-LLO-ISG15 than the PBS group (Appendix A), although statistical significance was not achieved for double and total cytokine-producing CD8^+^ T cells in both groups (Appendix A). Finally, we did not observe any significant difference in the myeloid-derived suppressor cells (MDSCs) population among treatments (Appendix A).

As vaccination with Lm-LLO-ISG15 resulted in robust cytokine-producing CD8^+^ T cell responses, we further investigated whether depleting CD8^+^ T cells could abrogate therapeutic efficacy of Lm-LLO-ISG15. Therefore, we conducted antibody-mediated depletion studies of CD8^+^ T cells in the MC38 tumor-bearing mice, as mentioned in the Materials and Methods section. We found that the tumor growth kinetics, as well as the final tumor mass, between the Lm-LLO-ISG15 + anti-CD8α and PBS-treated animals were similar (Figure 3M,N and Appendix A). This observation demonstrated the concept that Lm-LLO-ISG15-mediated efficacy in CRC is CD8^+^ T cell-dependent.

### 3.4. Induction of Anti-Tumor Immune Response with Lm-LLO-ISG15 by Oral and Intraperitoneal Administration in Orthotopic CRC Tumors

As previous studies have demonstrated the difference in response to immunotherapy between subcutaneous and orthotopic tumors [25,26], we aimed to evaluate whether the anti-CRC effects of Lm-LLO-ISG15 in subcutaneous tumors can also be translated to orthotopic tumors. In addition, we also wanted to investigate the potency of Lm-LLO-ISG15 when given orally versus intraperitoneally, as: (i) oral administration of Lm-based vaccines was shown to eradicate different types of tumors in earlier studies [27,28,29]; (ii) oral Lm vaccines are able to significantly induce antigen-specific T cell-mediated immune responses [29]; and (iii) the oral route of Lm-based vaccines offers ease of preparation and administration while reducing the cost of production compared with other vaccines formulated for injection [30].

We implanted luciferase expressing MC38 (MC38-GL) cell line intra-cecally to the colon of female C57BL/6 mice. The mice were then randomized prior to receiving PBS, Lm-LLO-ISG15 by i.p, or by oral gavage (p.o.), as described in Materials and Methods (Figure 4A). Consistent with the in subcutaneous models, Lm-LLO-ISG15 i.p significantly controlled the orthotopic-implanted MC38 tumors compared to PBS (Figure 4B,C and Appendix A). In contrast, while the oral administration of Lm-LLO-ISG15 also tended to control the tumor burden, we did not achieve statistical significance (Figure 4B,C). While there was no significant difference in the influx of total T cells among the three groups, we observed a trend of increased infiltration of CD4^+^ and CD8^+^ T cells in both Lm-LLO-ISG15 groups compared to PBS (Figure 4D–F). Noticeably, although Lm-LLO-ISG15 p.o tended to induce the highest influx of CD8^+^ T cells, this form of treatment also recruited the greatest number of Treg to the TME (Figure 4F,G). Consequently, the ratio of T_eff_/T_reg_ in the Lm-LLO-ISG15 p.o group tended to be the lowest among all three groups (Figure 4H). In contrast, while Lm-LLO-ISG15 i.p. recruited a higher number of CD8^+^ T cells to the TME compared to the PBS group, the Lm-LLO-ISG15 i.p. vaccination did not tend to attract the Treg population. As a result, the ratio of T_eff_/T_reg_ is the highest among all groups (Figure 4H). Despite the discrepancies in the infiltration of T cells between the two routes of Lm-LLO-ISG15 administration, CD8^+^ T cells from both the Lm-LLO-ISG15 i.p and p.o group were highly functional. This is demonstrated by a higher population of cytokine-producing CD8^+^ T cells in both treatment groups compared to that of PBS (Figure 4I–M).

The central role of CD8^+^ T cells in mediating the therapeutic efficacy of Lm-LLO-ISG15 in the orthotopic model was consistent with our observation in subcutaneous tumors. We did not observe any major difference in functional CD4^+^ T cells, i.e., single-, double-, and total-cytokine producing CD4^+^ T cells, among all of the groups (Appendix A). Interestingly, we did notice that triple-cytokine producing CD4^+^ T cells in Lm-LLO-ISG15 p.o were significantly higher compared to that of the Lm-LLO-ISG15 i.p. and PBS group. Similar to the subcutaneous models, we did not find any significant difference in the total MDSCs population, as well as monocytic MDSCs (m-MDSCs) and granulocytic MDSCs (g-MDSCs) subtype between the three groups (Appendix A). Collectively, we demonstrated that vaccination with Lm-LLO-ISG15 exerts an anti-tumor response in orthotopic CRC models. The delivery of Lm-LLO-ISG15 i.p. was associated with significant anti-tumor therapeutic efficacy, although the ability to generate an antigen-specific CD8^+^ T cell-mediated anti-tumor response between i.p. versus p.o was comparable.

### 3.5. Comparison of Efficacy of Lm-LLO-ISG15 versus Control Lm in Subcutaneous and Orthotopic CRC Mouse Models

To further validate our hypothesis that the therapeutic efficacy of Lm-LLO-ISG15 in CRC is mediated by the generation of an ISG15-specific anti-tumor immune response, we compared the anti-cancer potency of Lm-LLO-ISG15 with a Control Lm, i.e., a Listeria vaccine that does not secrete ISG15. We directly infected human and murine CRC cell lines in vitro with either Control Lm or Lm-LLO-ISG15. We observed a similarity in the direct cytolytic effect and invasive capacity in both vaccines in the human and murine CRC cell lines (Appendix A). Our current findings on the non-appreciable cytotoxic effects of Listeria-based vaccines were consistent with previous works in other cancer models [21]. It is well established now that the anti-tumor efficacy of Listeria-based vaccines is mainly dependent on the induction of anti-tumor immune responses.

Next, we treated MC38 tumor-bearing mice with either Control Lm or Lm-LLO-ISG15 and followed the animals for tumor growth kinetics in both subcutaneous and orthotopic models with experimental schema, depicted in Figure 5A,E, respectively. In the subcutaneous tumors, the mice that received Lm-LLO-ISG15 demonstrated a significantly reduced tumor growth rate (Figure 5B and Appendix A) and smaller final tumor mass (Figure 5C,D) compared to that of the control Lm. Similarly, treatment with Lm-LLO-ISG15 demonstrated better control over the tumor burden in the orthotopic CRC tumors (Figure 5F,G and Appendix A). The difference observed in the Lm-LLO-ISG15 versus Control Lm in vivo, but not in the in vitro studies, indicates that ISG15 provides a therapeutic target and is important to elicit an anti-tumor response in syngeneic CRC mouse models.

### 3.6. Induction of Anti-Tumor Immune Response with Lm-LLO-ISG15 in Subcutaneous and Orthotopic CRC Tumors

Next, to measure the generation of a tumor-specific T-cell response through systemic vaccination with Lm-LLO-ISG15 and the Control Lm, we performed intracellular cytokine staining in both the subcutaneous (Figure 6A–I) and orthotopic CRC models (Figure 6J–R).

In the subcutaneous tumors, the Lm-LLO-ISG15-treated mice demonstrated a marked increase in the infiltration of total T cells (Figure 6A), while the population of Tregs (Figure 6D) was significantly lower compared to that of the Control Lm-vaccinated mice. The ratio of T_eff_/T_reg_ in Lm-LLO-ISG15 was higher but did not reach statistical significance. The activated CD4^+^ T cells that produced IL-2 (Figure 6F,G), in particular, and all of the single cytokines (Appendix A) in general were significantly higher in the Lm-LLO-ISG15 group. In contrast to the striking difference in the CD8^+^ cells’ response between Lm-LLO-ISG15 and PBS, we did not observe any major difference in the influx nor the production of intracellular cytokines of CD8^+^ T cells in the subcutaneous MC38 between the Lm-LLO-ISG15 versus the control Lm cohort (Figure 6H,I and Appendix A). We also did not find any difference in other immune cell types in response to the Lm-based vaccines (Appendix A).

In the orthotopic model, the vaccination with Lm-LLO-ISG15 demonstrated a marked and significant increase in the infiltration of total T cells (Figure 6J), as well as CD4^+^, but not CD8^+^ T cells, compared to the Control Lm vaccination (Figure 6K,L). In addition, we noticed that the CD4^+^FoxP3^+^ T cell population was significantly lower in the Lm-LLO-ISG15-treated mice (Figure 6M). As a result, the T_eff_/T_reg_ ratio was significantly higher in the Lm-LLO-ISG15-vaccinated cohort. Consistent with the subcutaneous tumors, we found that activated the CD4^+^ T cells that produce IL-2 (Figure 6O) and all of the single cytokines (Appendix A) were significantly higher in the Lm-LLO-ISG15-vaccinated group. We observed a larger pool of IL-2-producing CD8^+^ T cells (Figure 6Q), as well as total cytokine-secreting CD8^+^ T cells (Appendix A). While there was a higher influx of m-MDSCs in the Lm-LLO-ISG15-vaccinated group, the Control Lm-vaccinated cohort was associated with an elevated g-MDSC population (Appendix A). Collectively, we demonstrated that therapeutic vaccination with Lm-LLO-ISG15 induced an effective ISG15-specific anti-tumor response mediated by T cells.

## 4. Discussion

While immunotherapy with ICIs in particular has demonstrated impressive results in mCRC patients with deficient mismatch repair (dMMR), the primary/acquired resistance and immune-related adverse effects (irAEs) associated with ICIs have established an unmet need to develop additional immunotherapy options for CRC patients [31]. Attenuated strains of *Lm* have been widely applied as therapeutic vaccines for the delivery and target of cancer antigens [13,21]. The findings presented in this manuscript supported the hypothesis that targeting TAA through active tumor immunotherapy such as in a *Lm*-based vaccine is a promising and novel therapeutic strategy in the treatment of CRC.

While ISG15 mRNA is almost undetectable in normal tissues (Appendix A), ISG15 is elevated in different types of cancers, including colon adenocarcinoma [32,33,34,35,36]. Our findings here suggest a strong correlation between the high expression of ISG15 and CRC progression and poor survival outcome of the disease (Figure 1). Therefore, we hypothesized that ISG15 could be considered as a TAA in CRC and potentially serve as a therapeutic target. We evaluated and observed the elevation of ISG15 in murine CRC tumors compared to the normal colon (Figure 2). We demonstrated, for the first time, as a proof of concept, that vaccination against ISG15 with Lm-LLO-ISG15 significantly controlled the CRC tumor burden in both subcutaneous and orthotopic syngeneic CRC mouse models (Figure 3 and Figure 4). In addition, we have shown that the anti-tumor efficacy of Lm-LLO-ISG15 was mediated by robust tumor-specific IFN-γ responses. Although CD4^+^ T cells might play a role in the anti-tumor response elicited by Lm-LLO-ISG15, as demonstrated by an increased influx of CD4^+^ T-cells in the TME, we found that the therapeutic efficacy of Lm-LLO-ISG15 is mainly mediated by the CD8^+^ T-cell. Recent studies have shown that the CD4^+^ T-cell also impacted the anti-tumor efficacy of other Listeria-based vaccines [37,38]. Therefore, it remains to be explored whether the depletion of CD4^+^ T-cells could abrogate the efficacy of Lm-LLO-ISG15. We further illustrated that treatment with Lm-LLO-ISG15 significantly enhanced the infiltration of functional T cells, and reduced the number of Tregs, thus increasing the ratio of T_eff_/T_reg._ Our results were consistent with previous studies evaluating the efficacy of Lm-LLO-ISG15 in breast cancer and renal cell carcinoma [39,40].

We assessed the difference in the therapeutic efficacy of Lm-LLO-ISG15 when delivered orally versus intraperitoneally. The scientific rationale for us to consider the oral delivery of Lm-LLO-ISG15 in our orthotopic CRC model is due to the possibility that the generation of mucosal immunity may provide greater therapeutic benefit in CRC. Previous studies also demonstrated that the oral administration of Listeria-based vaccines eradicated different types of tumors [27,28,29]. Further, the oral route of Listeria-based vaccines offers ease of preparation and administration compared with other formulations for injection. However, although the oral route of administration was employed with 10-fold higher bacterial doses, there were no additional anti-tumor benefits recorded. In fact, Lm-LLO-ISG15 i.p. was associated with a better control of the tumor burden (Figure 4B). Although increasing the dosage in the future, or using an alternative formulated form of delivery, such as encapsulation, can possibly enhance the therapeutic efficacy of Lm-based vaccines given orally, it is important to note that the oral delivery of *Lm*-based vaccines increases the population of Treg and leads to a lower T_eff_/T_reg_ ratio. While previous studies have reported tumor regression with the oral delivery of recombinant *Lm*-based vaccines [29], the mouse model was thought to be limited in the study of the oral delivery of Listeria-based vaccine due to the poor interaction between Internalin A and the host cell E-Cadherin receptor, thus limiting the entry of *Lm*-based vaccines [41]. Our findings in this manuscript confirmed this phenomenon again.

We also studied how Lm-LLO-ISG15 differed from Control Lm in controlling subcutaneous and orthotopic CRC mouse models (Figure 5 and Figure 6). We found a marked increase in the influx of total T cells in the Lm-LLO-ISG15 group compared to the Control Lm cohort. The influx of functional T cells, i.e., T cells that produce anti-tumor cytokines, including IL-2, IFN-γ, and TNF-α, in the TME of the Lm-LLO-ISG15-vaccinated group was higher than that of the Control Lm-treated group. Interestingly, the population of regulatory T cells was inversely lower in the Lm-LLO-ISG15-treated group. Consequently, this resulted in a higher ratio of T_eff_/T_reg_ in Lm-LLO-ISG15-vaccinated group as compared to that of the Control Lm group. We found that the Lm-LLO-ISG15 vaccine exerted better anti-tumor efficacy in the orthotopic, than subcutaneous, model of CRC. Previous studies have suggested that the TME of the orthotopic CRC model is considered “immunologically hot” compared to that of subcutaneous tumors [22,42]. This is because orthotopic CRC tumors were associated with a higher number of immune cell infiltrates, such as T cells and NK cells, while the number of MDSCs was lower compared to that of subcutaneous tumors [22,42]. As a result, the anti-tumor immune response in the orthotopic CRC models was relatively better than in the subcutaneous tumors [42]. Our findings presented in this manuscript were consistent with these reports. As orthotopic tumor-bearing mice have a longer median survival time, we were able to administer three doses of *Lm*-based vaccines instead of two doses in the subcutaneous models. Therefore, the orthotopic CRC models provided the *Lm*-based vaccines substantial time to recruit and activate the anti-tumor adaptive immune cells into the TME.

One of the major limitations of our current study is that we mainly studied the efficacy of the Lm-LLO-ISG15 vaccine in the MC38 model. MC38 represents the microsatellite instable (MSI) subtype of CRC tumors, which contributes to about 10–15% of the total CRC cases. The higher proportion of CRC, i.e., microsatellite stable (MSS), has not been discussed in the current manuscript. Further, we focused our investigation on understanding the therapeutic efficacy of Lm-LLO-ISG15 as a monotherapy in CRC. We did not compare and/or combine our vaccine with other currently approved therapies for CRC to enhance the final anti-tumor response. Another limitation of the current study is that we have not yet explored the protective potential of Lm-LLO-ISG15. In general, bacteria-based vaccines are thought to elicit robust protective immune responses and have demonstrated significant prophylactic efficacy in different cancer types [37,43]. Going forward, this could be another interesting avenue to be explored in future studies.

## 5. Conclusions

Collectively, we investigated and confirmed, for the first time, the therapeutic efficacy of Listeria-based vaccines targeting ISG15 (Lm-LLO-ISG15) in CRC. We found that Lm-LLO-ISG15 exerts an anti-tumor efficacy against syngeneic CRC mouse models and induces robust anti-tumor immune responses. Our results suggest Lm-LLO-ISG15 as a potential anti-cancer candidate for CRC treatment. Further investigations, such as combining Lm-LLO-ISG15 with other forms of immunotherapy and/or approved therapies, might enhance the overall anti-tumor efficacy of Lm-LLO-ISG15 and provide further support for the clinical translational promise of this therapy.

## Figures and Tables

**Figure 1 cancers-15-01237-f001:**
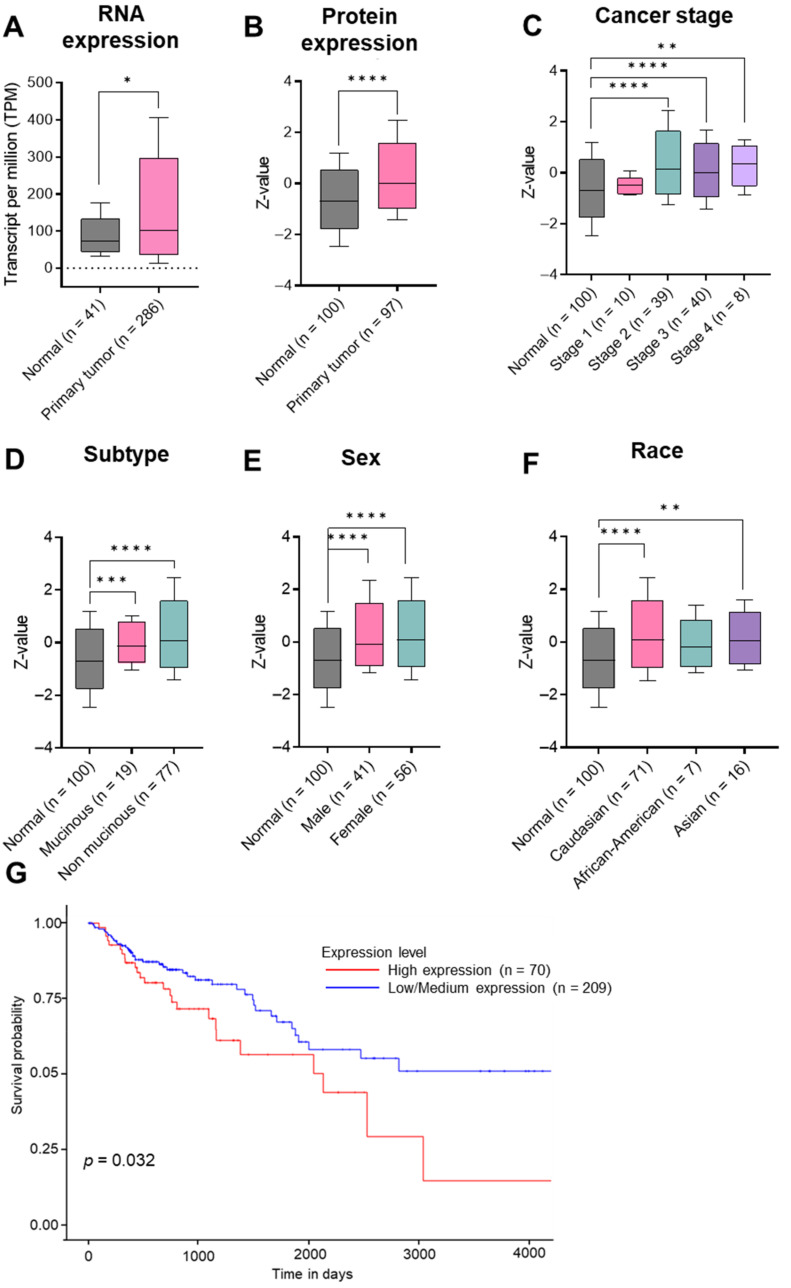
ISG15 expression and prognostic signature in human CRC. Gene expression (**A**) and protein expression (**B**) of ISG15 in normal colon tissues versus primary tumors in overall population. Protein level of ISG15 in different cancer stages (**C**), CRC subtypes (**D**), genders (**E**), and races (**F**). (**G**) Survival probabilities in CRC patients with high versus low/medium ISG15 mRNA expression. * *p* < 0.05, ** *p* < 0.01, *** *p* < 0.001, and **** *p* < 0.0001.

**Figure 2 cancers-15-01237-f002:**
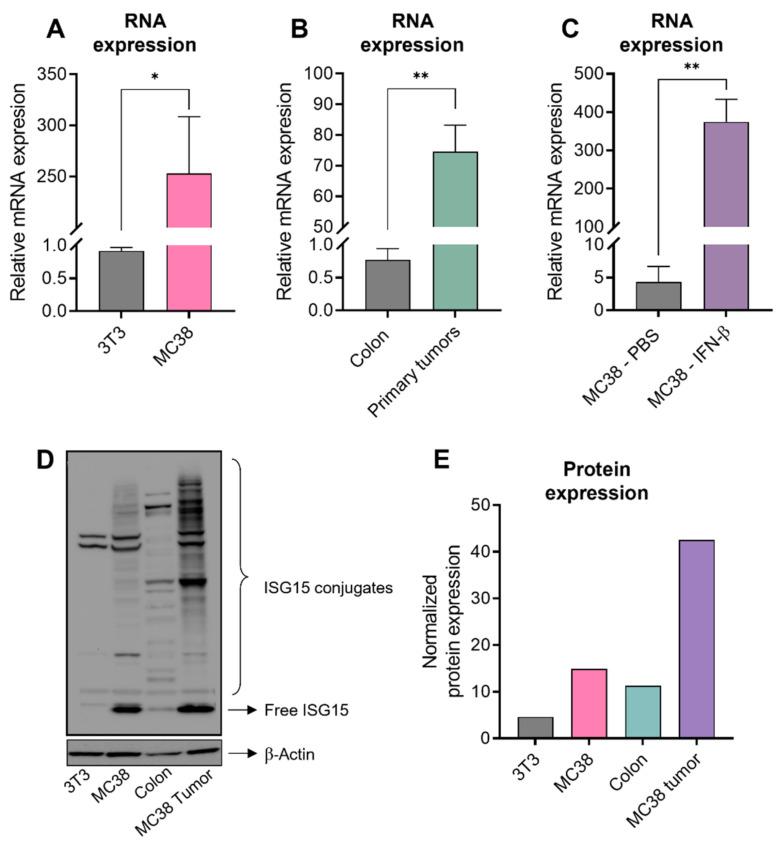
Elevation of ISG15 expression in murine CRC models. ISG15 mRNA expression relatively compared to 18S rRNA in (**A**) 3T3 vs. MC38 cell lines, and (**B**) colon tissues compared to MC38 subcutaneous tumors. (**C**) ISG15 is further induced in MC38 by 24 h exposure to IFN-β. (**D**) Western blot with (**E**) quantification demonstrating ISG15 expression and ISGylation in 3T3, MC38 cell line, colon tissues, and MC38 subcutaneous tumors. Data were analyzed using unpaired *t*-test. All error bars are shown as mean ± SEM. * *p* < 0.05, and ** *p* < 0.01.

**Figure 3 cancers-15-01237-f003:**
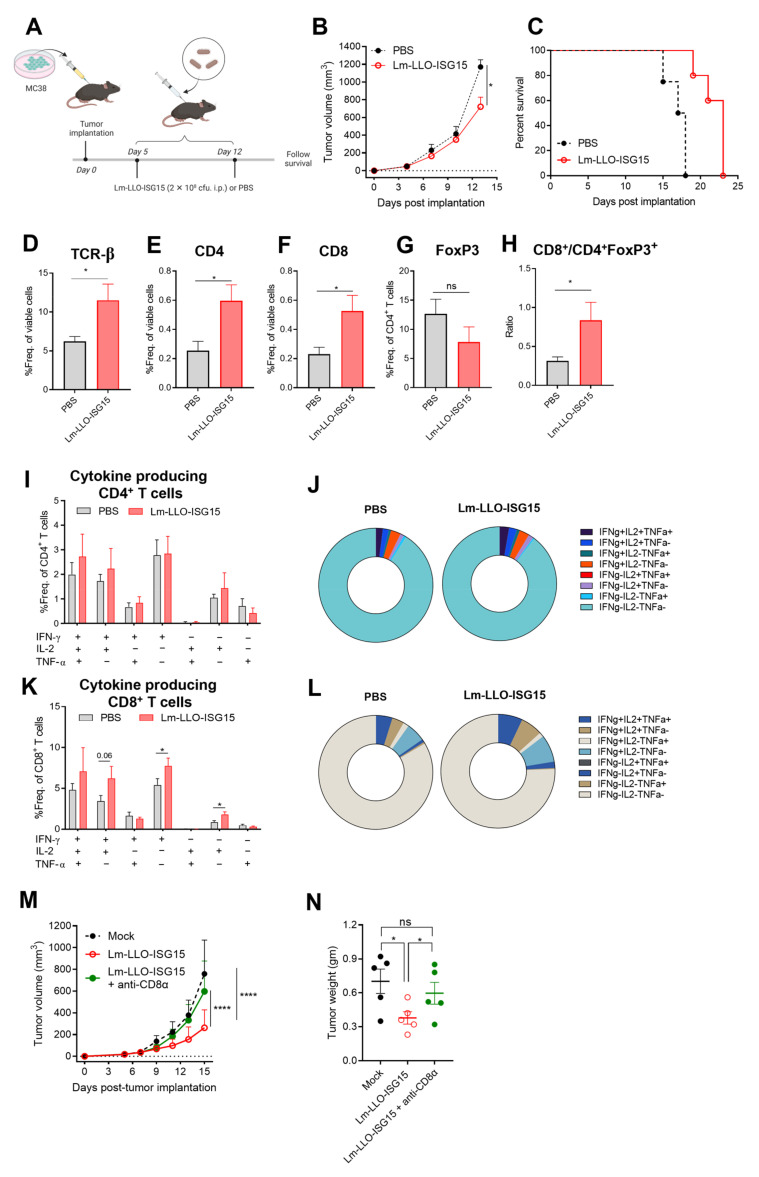
Lm-LLO-ISG15 vaccination results in anti-tumor efficacy in a subcutaneous CRC mouse model. (**A**) Experimental schema (made by BioRender). (**B**) Tumor growth kinetics (n = 4 in each group) and (**C**) survival probability of mice bearing subcutaneous MC38 tumors treated with PBS vs. Lm-LLO-ISG15 (17.5 vs. 23 days, respectively, *p* < 0.05). (**D**–**L**) Single cells from tumors were prepared for multicolor flow cytometry analysis as described in the Materials and methods section. Frequencies of live (**D**) TCR-β^+^, (**E**) CD4^+^, (**F**) CD8^+^, (**G**) CD4^+^FoxP3^+^, and (**H**) Ratio of CD8^+^% TCR-β^+^/CD4^+^FoxP3^+^% TCR-β^+^. (**I**,**J**) Cytokine-producing CD4^+^ T cells.(**K**,**L**) Cytokine-producing CD8^+^ T cells. (**M**) Tumor growth kinetics and (**N**) tumor mass of subcutaneous MC38 tumors treated with PBS, Lm-LLO-ISG15, or Lm-LLO-ISG15 + anti-CD8α. Data were analyzed using unpaired *t*-test. All error bars are shown as mean ± SEM. ns: non-significant, * *p* < 0.05 and **** *p* < 0.0001.

**Figure 4 cancers-15-01237-f004:**
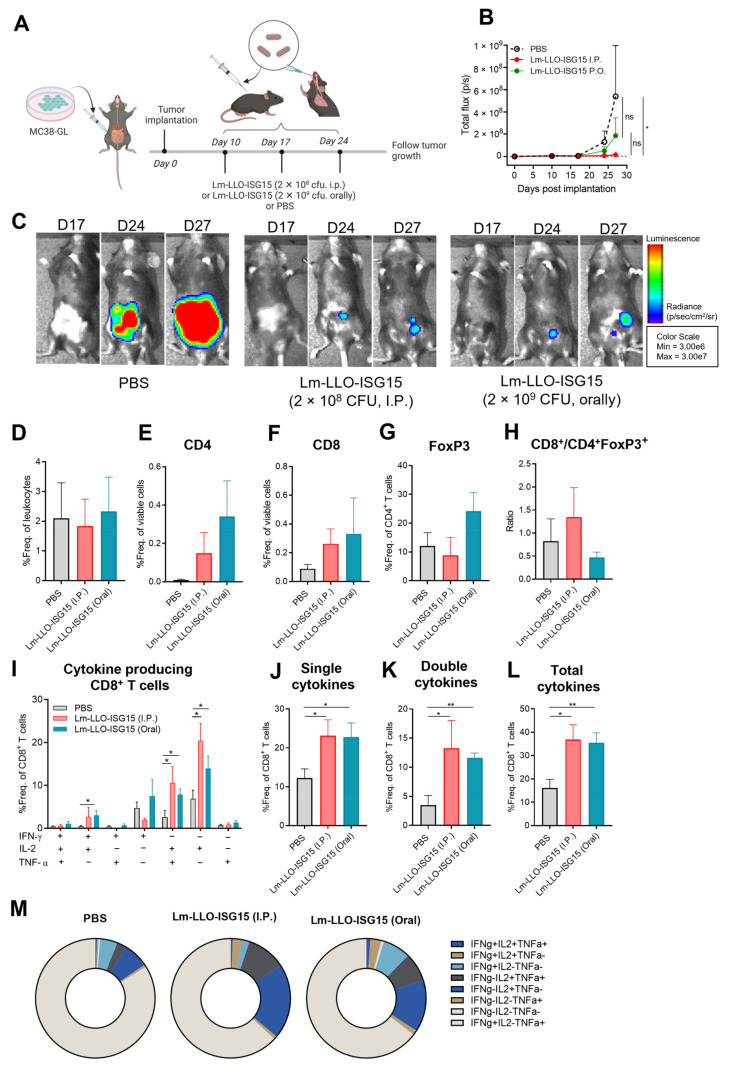
Efficacy of Lm-LLO-ISG15 i.p. versus p.o in orthotopic CRC mouse models. (**A**) Experimental schema (made by BioRender). (**B**) Tumor growth kinetics (n = 10 in each group) with (**C**) representatives of each group on day 17, 24, and 27. (**D**–**M**) Tumors were harvested to prepare single-cell suspensions and subjected to multicolor flow cytometry, as described in the Materials and methods section. Frequencies of live (**D**) TCR-β^+^, (**E**) CD4^+^, (**F**) CD8^+^, (**G**) CD4^+^FoxP3^+^, and (**H**) Ratio of CD8^+^% TCR-β^+^/CD4^+^FoxP3^+^% TCR-β^+^. (**I**) Multi-cytokine releasing CD8^+^ T cells. (**J**) Single cytokine producers in live CD8^+^ T cells. (**K**) Single cytokine producers in live CD8^+^ T cells. (**L**) Total cytokine producers in live CD8^+^ T cells. (**M**) Depiction of Distribution of multi-cytokine producers in live CD8^+^ T cells. Data were analyzed using unpaired *t*-test. All error bars are shown as mean ± SEM. ns: non-significant, * *p* < 0.05 and ** *p* < 0.01.

**Figure 5 cancers-15-01237-f005:**
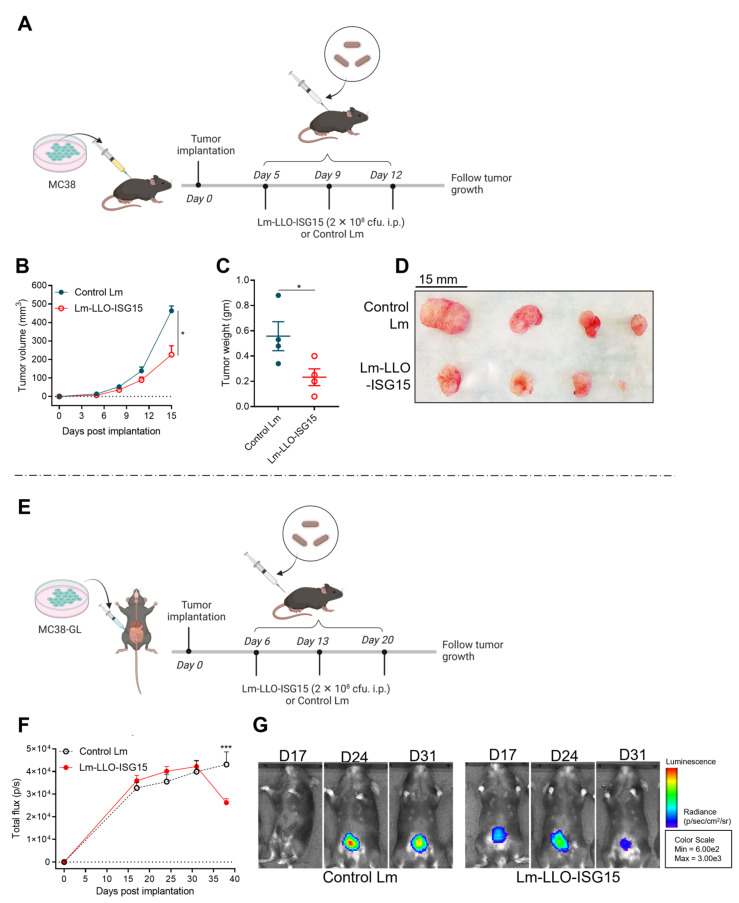
Therapeutic efficacy of Lm-LLO-ISG15 versus Control Lm in subcutaneous and orthotopic mouse models. (**A**) Experimental schema for subcutaneous model (made by BioRender). (**B**) Tumor growth kinetics (n = 4 in each group) and (**C**,**D**) final tumor mass. (**E**) Experimental schema for orthotopic model (made by BioRender). (**F**) Tumor growth kinetics (n = 8 in each group) and (**G**) representatives of tumor growth by bioluminesce signals on day 17, 24, and 31. Data were analyzed using unpaired *t*-test. All error bars are shown as mean ± SEM. * *p* < 0.05 and *** *p* < 0.001.

**Figure 6 cancers-15-01237-f006:**
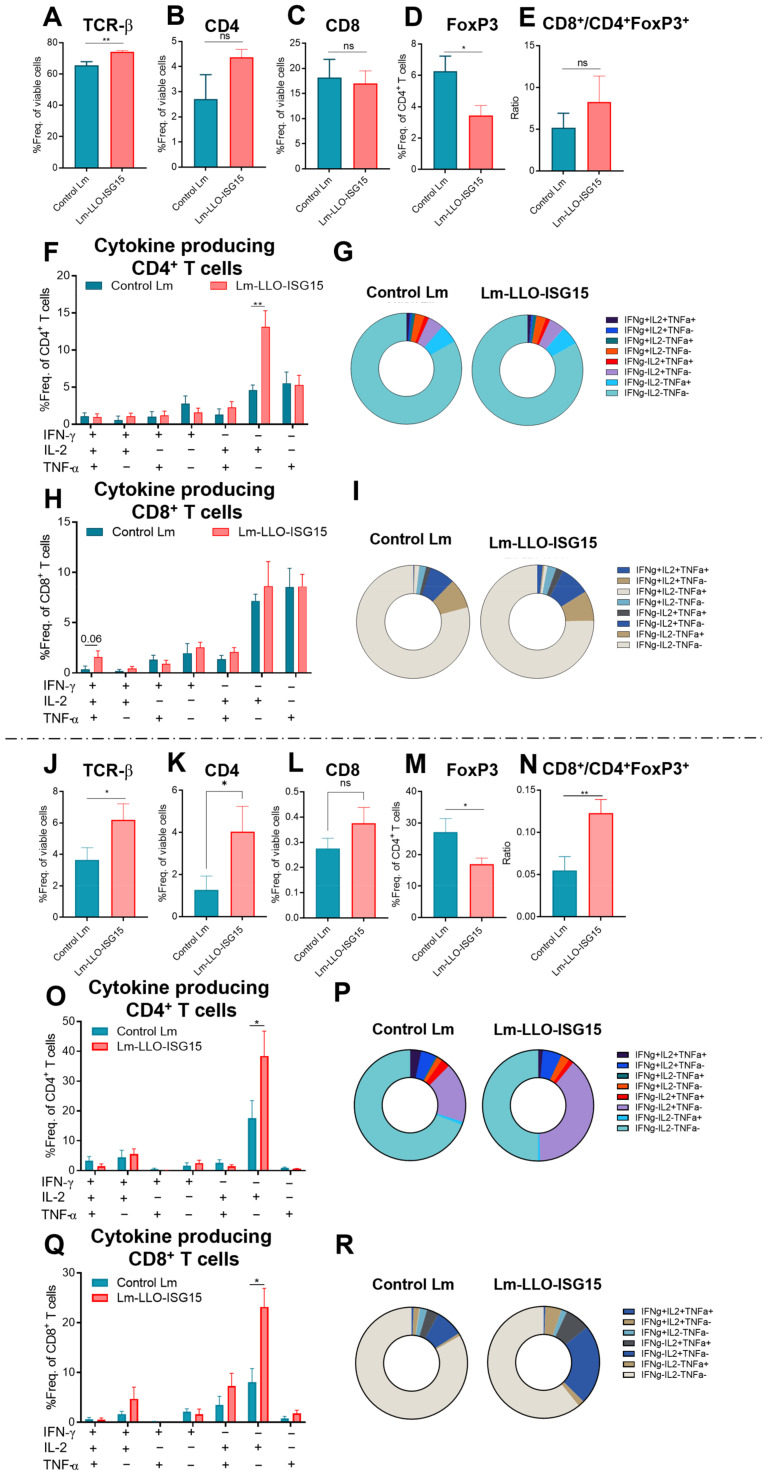
Induction of anti-tumor immune response with Lm-LLO-ISG15 in subcutaneous versus orthotopic CRC tumors. Single cells were prepared from tumors of experiment of Figure 5A and stained with different fluorochrome-conjugated antibodies as mentioned in Materials and methods. (**A**–**E**) Frequencies of live (**A**) TCR-β^+^, (**B**) CD4^+^, (**C**) CD8^+^, (**D**) CD4^+^FoxP3^+^, and (**E**) Ratio of Ratio of CD8^+^% TCR-β^+^/CD4^+^FoxP3^+^% TCR-β^+^. (**F**,**G**) Multi-cytokine produced by live CD4^+^ T cells. (**H**,**I**) Multi-cytokine produced by live CD8^+^ T cells. Similarly, single-cell suspensions of tumors from experiment of Figure 5E were subjected to multicolor flow cytometry. (**J**–**N**) Frequencies of live (**J**) TCR-β^+^, (**K**) CD4^+^, (**L**) CD8^+^, (**M**) CD4^+^FoxP3^+^, and (**N**) Ratio of Ratio of CD8^+^% TCR-β^+^/CD4^+^FoxP3^+^% TCR-β^+^. (**O**,**P**) Distribution of multi-cytokine produced by live CD4^+^ T cells. (**Q**,**R**) Distribution of multi-cytokine produced by live CD8^+^ T cells. Data were analyzed using unpaired *t*-test. All error bars are shown as mean ± SEM. ns: non-significant, * *p* < 0.05 and ** *p* < 0.01.

## Data Availability

Data on RNA and protein expression of ISG15 and the Kaplan–Meier survival analysis were obtained from Human Protein Atlas (https://www.proteinatlas.org/ (accessed on 17 November 2022)), The Cancer Genome Atlas (https://cancergenome.nih.gov/ (accessed on 17 November 2022)) and from the University of Alabama at Birmingham, Comprehensive Cancer Center (UALCAN, http://ualcan.path.uab.edu/ (accessed on 17 November 2022)) as previously published [44,45].

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
