# Peer review of "Targeting Ubiquitin-like Protein, ISG15, as a Novel Tumor Associated Antigen in Colorectal Cancer"

_cancers, 2023, doi:10.3390/cancers15041237_

Round 1

Reviewer 1 Report

This work by Nguyen et al. identified ISG15 as a potential tumor-associated antigen in human and murine CRC and developed a Lm-based vaccine targeting ISG15 for cancer treatment. The authors first discovered that ISG15 was upregulated in CRC tumors compared to normal tissues at mRNA and protein levels, which was conserved in the MC38 mouse CRC model. Then, the authors engineered the Lm-LLO-ISG15 vaccine to treat MC38 tumors inoculated subcutaneously and orthotopically and found that Lm-LLO-ISG15 treatment controlled tumor growth and increased T cell infiltration and functionality. The efficacy of vaccination by different administration routes was further examined. Mechanistically, the authors showed that the therapeutic effects of Lm-LLO-ISG15 were dependent on CD8+ T cells.

The study is overall interesting and provides valuable advancement in the field of cancer vaccination and immunotherapy. The experiments are properly designed, and references are nicely discussed. However, the flow of the manuscript is not optimal, and some results seem to be redundantly presented. Several experiments can be done to strengthen the manuscript. In addition, calculation errors can be found in the figures. These issues must be addressed before publication. Detailed major comments are listed below.

1. In the manuscript, the authors used PBS as a control in the initial subcutaneous and orthotopic experiments, where empty Lm or Lm expressing a different antigen should be used as another control for antigen specificity and innate immune response to Lm. In sections 3.5&6, the authors circled back to this question and compared the effects of a control Lm and Lm-LLO-ISG15. This structure makes the results look redundant and confuses the audience. It is suggested that the authors combine the results obtained from subcutaneous and orthotopic models in sections 3.5&6 into 3.3 and 3.4 then move some of the similar figures into the supplemental.

2. To prove the concept of tumor vaccination, the authors should assess the protective effect of Lm-LLO-ISG15 by vaccinating mice before tumor inoculation.

3.   Yadav et al. (Yadav et al. Nature, 2014) identified several classic neoepitopes presented by MC38 cells and showed that mice immunized with a pool of mutated peptides were substantially protected from tumor challenge. It would be interesting to compare the efficacy of vaccination with the neoepitopes and Lm-LLO-ISG15.

4. In the subcutaneous and orthotopic experiments, did any of the mice show complete tumor regression after Lm-LLO-ISG15 treatment? If so, it would be interesting to check whether the mice developed immunologic memory and were protected from rechallenge.

5.  In figures 4&6, it is not accurate to use the percentage of CD4 and CD8 cells relative to TCRb+ cells for the quantification of their infiltration. Frequencies relative to CD45+ cells or total live cells should be used instead.

6.  In all figures, the number of mice in each treatment group should be listed in the figure legends. Whether different independent experiments were pooled or a representative experiment was shown should also be described.

7. In figures 3&4&6, the ratio of CD8 T cells/Tregs was miscalculated. The frequency of Tregs in total T cells should be used for the calculation.

Other specific comments:

1.  The caption of figure 2E is missing.

2.  In figure 3B, how did the growth curves look after day 15? Was the difference bigger? Also, individual tumor growth curves should be included in the supplemental.

3.   The unit of tumor weight in figure 3N should be g or gm instead of mg.

4.    It is better to identify m-MDSCs and pmn-MDSCs using Ly6C and Ly6G.

5. The authors should include the detailed gating strategy for different cell populations in the supplemental.

6.  In figure 5F, why was the luciferase signal of the control Lm group much lower than that of the PBS group in figure 4B? Did control Lm treatment elicit any anti-tumor response? Also, did the difference get bigger after day 40 in figure 5F?

7.  In line 471, the word “warmer” is confusing and should be rephrased as immune hot. 

Reviewer 2 Report

The manuscript presented by Hong-My Nguyen et al descripted that Lm-LLO-ISG15 vaccination results in anti-tumor efficacy against the MC38 tumor model by recruiting Cytotoxic T Lymphocytes and leading to a more favorable effector to regulatory T cell ratio in the tumor microenvironment.

Recently many article papers mentioned that ISG15 is a novel tumor -associated antigen in tumor microenvironment. Such as Laurence Wood et al at 2015 shown that vaccination against ISG15 results in significant CD8-mediated reductions in both primary and metastatic mammary tumor burden (Cancer Immunol Immunother. 2012 May;61(5):689-700). This article seems to have the same theme with this manuscript.

What is a novelty of this manuscript. In addition, most experiments are lost control. Some conclusions are not support by evidence. Below are some comments to help strengthen the manuscript.

1.     Listeria-based vaccines are developed from an attenuated Listeria monocytogenes (Lm) bacterium which has the unique ability to preferentially infect antigen-presenting cells, such as dendritic cells (DCs), to deliver the antigen of interest and induce robust cytotoxic T lymphocyte (CTL) responses that eradicate tumors. The authors should show ISG15 expression in DC cells is correlated with prognostic signature.

2.     For Figure 3B-L, The control cannot not design with PBS, should be Lm.

3.     For Figure 3M, there did not have IgG and anti-CD8a alone as control.

4.     The authors observed an increased influx of total T cells as well as CD4+ T cell and CD8+ T cell subsets in the Lm-LLO-ISG15 group compared to that of PBS (Figure 3D-F).  they need discuss whether depleting CD4+ T cells could abrogate therapeutic efficacy of Lm-LLO-ISG15.

5.     For Figure 4, The control should be Lm, not PBS. The authors used for 3 mice in each group. It is better to do with at least 5 mice in each group.

6.     For Figure 5F, the tumor grown curves are strange. The size of the tumor suddenly decreased. It is better to do with at least 5 mice in each group.

7.     Figure Legends are way too vague. Such as no Statistical analysis data...

Reviewer 3 Report

Hong-My Nguyen et al.’s manuscript “Targeting ubiquitin-like protein, ISG15, as a novel tumor 2 associated antigen in colorectal cancer” developed a Listeria-based vaccine (i.e., Lm-LLO-ISG15) that targets the interferon-stimulated gene 15 (ISG15), and demonstrated its anti-tumor efficacy in an immunocompetent CRC murine model (MC38). Overall, it is an interesting topic that might interest people in this area greatly. However, there are some major concerns and questions need to addressed.

1.     Significance of data. In Fig 1B, Fig 4B and Fig 5F, the Lm-LLO-ISG15 somehow showed some tumor regression efficacy, however, the effect was marginal and only showed difference at the endpoint. Especially the Fig 5F, how would the authors explain the sudden decrease at day 38? Why didn’t the authors extend the studies to see if the tumor regression efficacy of Lm-LLO-ISG15 can be continued and further differentiate the treatment vs disease control groups?  The error bar of Fig 4B was huge, how many mice were used in the study?

2.     The authors showed that ISG15 was expressed higher in cancer patient as well as murine tumor model. However, as an antigen target, the specificity is important. There was no information in the manuscript to show which cells expressed the ISG15. Also, they did not show the ISG15 expression of healthy mice/tissues. How specific is the target? What’s the expression level of ISG15 in healthy mice/tissue?

3.     As illustrated in the graphical abstract, the proposed mechanism of the vaccine Lm-LLO-ISG15 was to stimulate a response from the immune system and thus, create a 'memory’ in the immune system. However, in all the in vivo studies, Lm-LLO-ISG15 was administrated after tumor implantation. Have the authors tried to immunize the mice with Lm-LLO-ISG15 prior to tumor implant and see if it developed the immunological memory? Besides, the authors should test the if the mice produced antibody or cell immunity against the antigen ISG15 to validate the mechanism.   

Reviewer 4 Report

Can you add the primers used for the PCR and what was the control. 

For FACS did you pay attention to the compensation? Antibodies alone? 

Round 2

Reviewer 1 Report

The authors addressed most of the reviewers’ comments and better discussed the limitations of the current work. Now the manuscript is suitable for publication. As also suggested by another reviewer, vaccination with Lm-LLO-ISG15 before tumor challenge is interesting and should be accessed in the authors’ future study. However, there are still minor changes that need to be done in the current manuscript.

1.       The authors used the ratio CD8+%TCRb+/Foxp3+%CD4+ for calculation as CD8+/Foxp3+ was found to be a better prognosticator in the cited paper. However, the authors should note that the analysis was done on patient IHC images, which can not be applied to flow. In the same paper, a ratio of CD3+CD8+% CD45+/ CD3+CD4+CD25+Foxp+% CD45+ using flow was also examined for its prognostic value. In addition, this method has been used in numerous papers (PMID: 36169564, PMID: 29669930, PMID: 31350404) and is widely accepted in the field of immunology. The authors should correct the calculations in all figures as this may lead to misinterpretation of the data.

2.       The authors should change the description in line 421 to reflect the change in figure 6K&L. 

Author Response

Reviewer 1:

The authors addressed most of the reviewers’ comments and better discussed the limitations of the current work. Now the manuscript is suitable for publication. As also suggested by another reviewer, vaccination with Lm-LLO-ISG15 before tumor challenge is interesting and should be accessed in the authors’ future study.

Response: Thank you. We appreciate positive feedbacks.

However, there are still minor changes that need to be done in the current manuscript.

  1. The authors used the ratio CD8+%TCRb+/Foxp3+%CD4+ for calculation as CD8+/Foxp3+ was found to be a better prognosticator in the cited paper. However, the authors should note that the analysis was done on patient IHC images, which cannot be applied to flow. In the same paper, a ratio of CD3+CD8+% CD45+/CD3+CD4+CD25+Foxp+% CD45+using flow was also examined for its prognostic value. In addition, this method has been used in numerous papers (PMID: 36169564, PMID: 29669930, PMID: 31350404) and is widely accepted in the field of immunology. The authors should correct the calculations in all figures as this may lead to misinterpretation of the data.

Response: Thank you. While using CD8+%TCR-β+/CD4+Foxp3+%TCR-β+, we agreed that CD3+CD8+%CD45+/CD3+CD4+CD25+Foxp+%CD45+ is widely used in the field of immunology. However, in place of CD45+CD3+, TCR-β+ is also commonly used to identify and detect CD4+ and CD8+ T cells (PMID: 34597344, PMID: 35523960, PMID: 33572870). Unfortunately, we did not include CD25+, CD3+, and CD45+ in our flow analysis panel and we did not have those samples left to re-run the flow analysis. We clarified about the Teff/Treg ratio in our figure legend. We appreciate your suggestion and will take deliberately consideration regarding this matter in our future studies.

  1. The authors should change the description in line 421 to reflect the change in figure 6K&L.

Response: Much appreciated. We changed as you suggested.

Reviewer 2 Report

The authors addressed most of questions.

Author Response

We thank you for your time and effort to provide constructive comments and suggestions that have improved our manuscript.

Reviewer 3 Report

The authors have addressed my questions and concerns. I recommend the manuscript to be published after checking the figures and text thoroughly.  

Author Response

(The authors gave the same response as above.)
